# Interactions between Cytokines and the Pathogenesis of Prion Diseases: Insights and Implications

**DOI:** 10.3390/brainsci14050413

**Published:** 2024-04-23

**Authors:** Gabriela Assis-de-Lemos, Rayanne Moura-do-Nascimento, Manuela Amaral-do-Nascimento, Ana C. Miceli, Tuane C. R. G. Vieira

**Affiliations:** Institute of Medical Biochemistry Leopoldo de Meis and National Institute of Science and Technology for Structural Biology and Bioimaging, Federal University of Rio de Janeiro, Rio de Janeiro 21941-902, RJ, Brazil; gassis@bioqmed.ufrj.br (G.A.-d.-L.); rayannemourabio@gmail.com (R.M.-d.-N.); manuelaamaral11@gmail.com (M.A.-d.-N.); anacarolinamiceli@gmail.com (A.C.M.)

**Keywords:** neuroinflammation, cytokine modulation, prion diseases

## Abstract

Transmissible Spongiform Encephalopathies (TSEs), including prion diseases such as Bovine Spongiform Encephalopathy (Mad Cow Disease) and variant Creutzfeldt–Jakob Disease, pose unique challenges to the scientific and medical communities due to their infectious nature, neurodegenerative effects, and the absence of a cure. Central to the progression of TSEs is the conversion of the normal cellular prion protein (PrPC) into its infectious scrapie form (PrPSc), leading to neurodegeneration through a complex interplay involving the immune system. This review elucidates the current understanding of the immune response in prion diseases, emphasizing the dual role of the immune system in both propagating and mitigating the disease through mechanisms such as glial activation, cytokine release, and blood–brain barrier dynamics. We highlight the differential cytokine profiles associated with various prion strains and stages of disease, pointing towards the potential for cytokines as biomarkers and therapeutic targets. Immunomodulatory strategies are discussed as promising avenues for mitigating neuroinflammation and delaying disease progression. This comprehensive examination of the immune response in TSEs not only advances our understanding of these enigmatic diseases but also sheds light on broader neuroinflammatory processes, offering hope for future therapeutic interventions.

## 1. Introduction

Prion diseases, also known as Transmissible Spongiform Encephalopathies (TSEs), constitute a unique class of neurodegenerative disorders that are both rare and infectious. These diseases have captivated the scientific community due to their distinct characteristics and the absence of a cure. The central mechanism underlying TSEs involves the structural conversion of the cellular prion protein (PrPC) into a pathological isoform, prion scrapie (PrPSc) [1]. This conversion marks a pivotal event in the progression of TSEs. Notably, TSEs exhibit infectious properties, with evidence suggesting the transmission of PrPSc across species barriers, a phenomenon that challenges traditional understandings of infectious diseases [2]. The term “prion” itself is derived from a contraction of “protein” and “infection”, aptly denoting its nature as a proteinaceous infectious particle.

Bovine Spongiform Encephalopathy, more commonly called Mad Cow disease, is an example of prion disease that garnered global attention [3]. The vertical transmission of the pathological PrPSc via consuming contaminated meat precipitated an outbreak of variant Creutzfeldt–Jakob Disease (vCJD), marking a significant public health crisis in the 1980s [4,5]. In addition to vCJD, the spectrum of human prion diseases encompasses Gerstmann–Sträussler–Scheinker Syndrome (GSS), Fatal Familial Insomnia (FFI), and Kuru, each characterized by distinct clinical and neuropathological features [4,5]. Historically, the infectious nature of prion diseases led to the hypothesis that their etiological agent was a slow virus, noted for its prolonged incubation period before the onset of clinical symptoms. This perspective persisted until the unique properties of prions as proteinaceous infectious particles were elucidated, challenging conventional paradigms of infectious disease [1].

PrPSc originates from the self-propagating alteration of PrPC through conformational remodeling, wherein PrPSc induces the misfolding of PrPC via a template-directed mechanism. Acting as a template, PrPSc incorporates PrPC into amyloid aggregates, ultimately leading to neuronal death [6]. Given that PrPSc and PrPC share identical amino acid sequences, they are inherently recognized as endogenous by the host’s immune system, which consequently exhibits natural tolerance towards them. This intrinsic similarity appears to preclude the activation of an adaptive immune response following prion infection [7,8,9]. Nonetheless, the immune system plays a pivotal albeit complex role in the pathogenesis of prion diseases, characterized by a dual functionality: initially, post-infection, prion targeting to lymphoid tissues is essential for the disease’s propagation to the central nervous system; subsequently, within the brain, the activation of microglia and their resultant cytokine production not only exacerbates neuroinflammation and brain damage but may also mitigate the start of prion diseases by phagocytosing and degrading PrPSc particles [7].

To date, the intricacies of immunologic responses activated in prion diseases remain incompletely understood. In this review, we unravel the intricate interplay between prions and the host’s immune response. We progress from detailing the immune response in prion infections to analyzing the roles of cytokines in prion pathogenesis, emphasizing their dualistic nature in mediating neuroprotection and neuroinflammation. Then, we make a critical analysis of neuroinflammatory cascades, highlighting the contributions of various immune cells to the disease process. Finally, we contemplate emerging immunotherapeutic strategies, pointing toward future research directions. Unraveling the mechanisms of neuroinflammation in prion disease not only sheds light on their unique pathogenic processes but also enhances our understanding of other neurodegenerative disorders. By exploring the interplay between prions and the immune system, we aim to illuminate the complex biological responses contributing to the progression and manifestation of TSEs.

## 2. Immune Response Activation in Prion Infection

The immune system exhibits a dual role in the context of prion diseases. In the initial phases, it mobilizes to detect and eliminate prions, functioning as a critical defense mechanism. Yet, as the disease advances, this immune response becomes dysregulated, culminating in excessive inflammation and subsequent neurodegeneration. Understanding the complex interplay between prions and the immune system is pivotal for devising strategies to modulate these responses effectively, with the aim to mitigate the adverse effects while enhancing protective mechanisms.

### 2.1. Glia Activation

Although neurons are the cell type most affected in TSEs, as they ultimately die, the activation of glial cells such as astrocytes and microglia is an important component of the physiopathology of these diseases which has been increasingly characterized more recently [10,11,12,13,14]. In mice infected with prions, disease progression is marked by notable changes in glia-enriched genes that align with the emergence of clinical symptoms. Interestingly, these alterations precede significant neuronal changes, which become evident primarily at the terminal stage of the disease [13]. Furthermore, a cell-type specific ribosome profiling analysis revealed that most alterations in mRNA translation throughout the progression of prion disease are observed in non-neuronal cells. Neurons, in contrast, exhibit only minimal changes in their translational profile [15]. These studies suggest that alterations in glial function could represent an early event in the pathogenesis of prion diseases, initiating a cascade of events that ultimately lead to neuronal damage. Conversely, the expression of PrPSc in cell types other than neurons appears insufficient to induce glial activation and subsequent neurodegeneration in transgenic mice models [11]. Notably, neurodegeneration was observed only when PrPSc was propagated in neuronal cells, and PrPSc from astrocytes did not induce microgliosis or astrogliosis [11]. The presence of PrPC in microglia for its activation is controversial, with studies showing its non-essential role in LPS-induced activation [16] and its contradictory necessity [17], a discrepancy that may stem from differences in the genetic backgrounds of the knockout models used. Indeed, analysis of tissue samples from patients afflicted with FFI and genetic CJD (gCJD) revealed an absence of microglia activation, a response observed only in cases of sporadic CJD (sCJD) [12]. Even in sCJD, microglial activation depends on the type of PrPSc [18]. Microglia activation was also observed in GSS cases [19] and after prion infection [11,20]. This distinction emphasizes the complex nature of brain degeneration in prion diseases, indicating unique immune response mechanisms in different prion diseases. The accumulation of PrPSc in neurons may trigger an initial activation of glial cells, setting off a series of events that ultimately lead to neuronal death. However, this glial activation might not be evident in the advanced stages of the disease, suggesting a temporal and potentially disease-specific pattern of immune response within the central nervous system (Figure 1).

Research into the involvement of glial cells in the pathophysiology of TSEs highlights a dual, yet ambiguous, role of glial activation: it can be both protective and detrimental. In response to a pathogen or a foreign entity, activated glial cells orchestrate a complex immune response by producing anti-inflammatory and pro-inflammatory cytokines, thus engendering an inflammatory milieu within the central nervous system (Table 1).

Interestingly, a deficiency in microglial activity has been linked to the accelerated progression of prion diseases, potentially associated with PrPSc deposition in the brain [16,33,34]. This observation is complemented by findings that the translational profile characteristic of microglia deteriorates progressively across the stages of prion disease [35], suggesting a protective role for glial cells in TSEs. This protective aspect is further highlighted by the role of the anti-inflammatory cytokine IL-10, predominantly released by astrocytes and microglia in response to an insult [21]. IL-10 deficiency accelerates prion disease, evidenced by a reduced incubation period [36,37].

Conversely, the deposition of PrPSc within neurons has been shown to trigger microglial activation [11]. Within the neuroinflammation context, microglial activation, through the release of pro-inflammatory cytokines such as IL-1α, TNF-α, and C1q, promotes the transformation of astrocytes into the A1 neurotoxic phenotype. These A1 astrocytes produce chemokine (C-C motif) ligand 2 (CCL2), a potent chemoattractant that recruits immune cells to the central nervous system, exacerbating the disease state [22,24,38]. CCL2, secreted by reactive A1 astrocytes in response to TNF-α exposure, triggers the transformation of microglia into the neurotoxic M1 phenotype. This conversion was not observed when CCL2’s action was inhibited or when its expression was suppressed through siRNA techniques [25]. M1 microglia releases inflammatory mediators that exacerbate neuroinflammation and neurotoxicity [39]. This leads to the further production of A1 astrocytes, creating a feedback loop of inflammation that may ultimately result in neuronal damage [25]. Since A1 astrocytes are prevalent in prion diseases [38], in conjunction with its upregulation being observed in brains affected by sporadic CJD [40] and in other major neurodegenerative diseases [22], it is possible to hypothesize that an inflammation response can happen in prion diseases (Figure 2). However, there are many open questions since the removal of TNF-α, IL-1α, and C1qa had a minimal impact on suppressing A1-specific markers in prion-infected animals, which unexpectedly hastened the progression of prion diseases [40]. This finding suggests that astrocyte activation in prion diseases may occur independently of microglia. Furthermore, there is also an increase in A2 astrocytes in prion-infected mice [41,42], suggesting a simultaneous neurotoxic and neuroprotective response during infection and the subsequent dysregulation of these responses concerning neurodegeneration.

In the context of prion diseases, glial activation could represent an initial defense mechanism against the accumulation of PrPSc, given that microglia naturally phagocytize abnormal proteins in the brain. However, their inability to effectively remove PrPSc [43] might trigger chronic brain inflammation. This is due to an overactive immune response, potentially contributing to neuronal damage in TSEs [44] (Figure 2).

### 2.2. Neuronal Response

Investigations into neuronal cytokine production challenge the conventional view that glial cells are the predominant source of cytokines within the brain. To date, evidence specifically documenting the neuronal release of cytokines in the context of TSEs or other neurodegenerative diseases remains elusive. Nonetheless, there has been a demonstration of cytokine release from neurons triggered by the in vitro activation of P2X7 receptors, a mechanism known to facilitate cytokine activation in other cell types [23]. IL-3, prominently released by neurons alongside other cytokines like TNF-α, IL-4, and IL-10, has been shown to protect neurons from death experienced over time in culture conditions [23]. This finding implies that neurons may engage in an active role within the cytokine-mediated communication relevant to neurodegeneration, employing autocrine signaling for self-protection.

Historically, detecting immune cytokines within the central nervous system led to the supposition that these bioactive molecules were endogenously synthesized due to the assumed restrictive nature of the blood–brain barrier (BBB). This perspective posited that the brain autonomously produced cytokines in situ. Subsequent research, however, has refined this view. It has been shown that the BBB, while selectively permeable, does permit the passage of specific peripheral cytokines including IL-1α and IL-6 into the brain parenchyma, challenging the previous understanding of cytokine sources [45].

In addition to glial cells, neurons have been demonstrated to express a diverse array of cytokines [46,47,48]. However, the release of these cytokines into the extracellular medium remains significantly less investigated. The first studies into the physiological expression of immune cytokines by neurons, distinct from expressions linked to illness, utilized in situ hybridization techniques to reveal: (i) the detection of IL-6 mRNA and its colocalization with the IL-6 receptor (IL-6R) in neurons across various brain regions [49,50]; (ii) in cultured cortical neurons, an upregulation of IL-6 expression triggered by direct exposure to proinflammatory cytokines IL-1 and TNF-α [26]; (iii) the presence of IL-1β with the axons of human hypothalamic neurons [51]. Together, these studies point to the physiological importance of cytokines in neuronal communication. Consequently, given that the neuronal expression of IL-6 can be triggered by pro-inflammatory stimuli in the brain [26], it stands to reason that the inflammation via glial activation associated with neurodegeneration may lead to an upregulation of inflammatory cytokines such as IL-6 and TNF-α in neurons as an adaptative mechanism (Figure 3). This upregulation could affect neuronal dynamics, for example, by altering the expression of glutamate receptors and L-type calcium channels [52,53,54,55], showcasing the intricate relationship between inflammatory processes and neuronal activity.

Peripheral infections can also influence the expression of immune cytokines within the brain. For instance, TNF-α levels increased in hypothalamic neurons following systemic administration of bacterial lipopolysaccharide (LPS) in mice [56]. A more recent investigation into the impact of peripheral inflammation on brain cytokine levels utilized carrageenan-induced hind paw edema in rats. This study revealed a significant upregulation of Scya2 mRNA, which encodes the chemokine Monocyte chemotactic protein-1 (MCP-1) within a distinct population of neurons [28]. Peripheral inflammation has been identified as a contributing factor to the progression of neurodegenerative diseases [57,58]. This relationship may stem from an intensified inflammatory response within the brain, due to the upregulation of neuronal cytokines (Figure 3).

During infection, neurons can express signaling molecules that attract microglia. These microglia, in turn, release cytokines and chemokines, drawing additional immune cells to the infection site [59]. It is reasonable to suggest that in the context of prion diseases, neurons might similarly recruit immune cells in reaction to the presence of PrPSc, leading to a localized increase in cytokine release. On the contrary, neurons are also known to express molecules that signal a decrease in local inflammation. For example, CD22, a ligand for the CD45 receptor, when expressed by neurons, inhibits the microglia production of pro-inflammatory cytokines [60] via the activation of the CD45 receptor, which has been shown to downregulate microglial activation in response to β-amyloid peptide [61]. Consequently, neuronal signaling molecules are critical in modulating the brain’s inflammatory status during prion diseases by influencing cytokine release from glial cells (Figure 3).

**Figure 3 brainsci-14-00413-f003:**
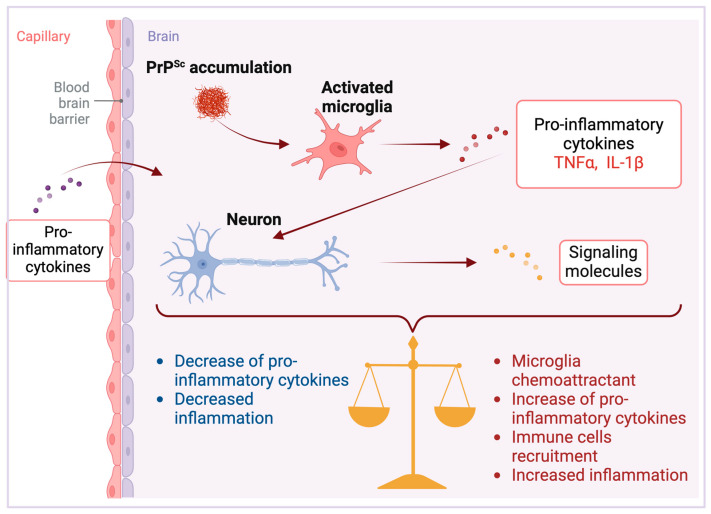
Neuronal responses in prion diseases. PrPSc accumulation triggers microglial activation [11], producing pro-inflammatory cytokines such as tumor necrosis factor-alpha (TNFα) and interleukin-1 beta (IL-1β). Beyond endogenous production, these cytokines can also permeate the brain parenchyma from peripheral tissues [45]. Once within the central nervous system, they induce neurons to release signaling molecules with the potential to both escalate and mitigate inflammation [28,57,58]. These neuronal signals can propagate pro-inflammatory effects by attracting further microglial and immune cell engagement, intensifying neuroinflammation [59]. In contrast, some molecules initiate anti-inflammatory mechanisms that suppress pro-inflammatory cytokine activity, thereby reducing inflammation [60,61]. The equilibrium between these antagonistic signals governs the inflammatory state of the brain, establishing a critical balance that dictates the establishment and progression of disease pathology. Created with BioRender.com (accessed on 16 April 2024).

### 2.3. The Blood–Brain Barrier

The BBB is structured by endothelial cells interacting with pericytes, neurons, astrocytes, and microglia. This barrier is essential for maintaining the homeostatic microenvironment of the central nervous system. Physiologically, the BBB controls the trafficking of immune cells to the brain, which is restricted in a healthy brain [62]. However, disruptions in the integrity and permeability of the BBB can result in the dysregulated trafficking of immune cells into the brain, thereby promoting neuroinflammation [63].

Inflammation associated with neurodegeneration in multiple sclerosis (MS) and experimental autoimmune encephalomyelitis (EAE) was shown to disrupt BBB tight junctions (TJ) [64]. Inflammatory cytokines such as IL-17 and IL-22 secreted by Th1 and Th17 lymphocytes may lead to the downregulation of occludin, a component of TJ strands, in a mechanism dependent on reactive oxygen species (ROS) [29]. In the acute phase of MS and EAE, CCL2 is elevated (besides TNF-α and IL-1β), and this can be related to BBB dysfunction since this cytokine was shown to promote the degradation of claudin-5, the main component of TJ [65]. As PrPSc deposition leads to the production of CCL2 by A1 astrocytes [22,38] (Figure 2), it may also lead to BBB dysfunction in prion diseases.

Recent studies have demonstrated that in prion-infected mice, the integrity of the BBB is compromised prior to the manifestation of clinical symptoms [27]. Furthermore, endothelial cells isolated from these mice exhibit reduced levels of the tight junction proteins occluding and claudin-5, as well as the adherens junction protein VE-cadherin [27]. Interestingly, this compromise in barrier integrity could be attributed to high levels of IL-6 secreted by reactive astrocytes. Such an increase in IL-6 was observed to reduce the levels of TJ proteins in non-infected mice-isolated endothelial cells within a co-culture model [27], although IL-6 itself is not directly linked to prion diseases progression [66]. Also, TNF-α has been shown to promote a decrease in claudin-5 expression in mice-isolated endothelial cells [67], which may suggest that this cytokine contributes to BBB dysfunction in TSEs when released by activated microglia in response to PrPSc [22].

In Alzheimer’s disease, which exhibits pathological hallmarks akin to those observed in TSEs [68,69], postmortem brain analyses have revealed a pronounced loss of TJ proteins, specifically claudin-5, occluding, and Zonula occludens-1 (ZO-1). This loss is notably correlated with the accumulation of β-Amyloid (AB) in cortical capillaries [70]. This accumulation leads to the activation of microglia, which release pro-inflammatory cytokines and also ROS, leading to the disruption of BBB integrity through the loss of TJ proteins [70].

While the direct impact of cytokine signaling on BBB function during neurodegenerative processes remains poorly understood, evidence from other neuropathological conditions points to BBB dysfunction because of the release of inflammatory cytokines [63,71]. Microglial-released TNF-α was the primary mediator of the necroptosis of endothelial cells in a cerebral ischemic/reperfusion injury model, contributing to the disruption of BBB selectivity [72]. Mechanistically, a previous study demonstrated that TNF-α induces an increase in the phosphorylation of occludin through MAP kinases in a human endothelial cell line culture. After prolonged exposure, there is a subsequent decrease in occludin expression. Accompanying these molecular alterations are morphological changes in the cells and increased intercellular permeability. These findings suggest a potential pathway by which TNF-α may contribute to BBB disruption during neuroinflammation [73].

Interestingly, the pro-inflammatory cytokine IL-1β may influence endothelial cell function via a distinct mechanism. It has been demonstrated that IL-1β stimulates endothelial cells to upregulate hypoxia-inducible factor-1alpha (HIF-1α) and vascular endothelial growth factor (VEGF). This response triggers the hypoxia-angiogenesis pathway, leading to the activation of multiple pro-angiogenic pathways and, consequently, heightened BBB permeability [74]. Furthermore, within an in vitro human THBMEC-based blood–brain barrier model, IL-1β was found to significantly upregulate the mRNA expression of the pro-inflammatory cytokines IL-6, IL-8, and TNFα in the endothelial cell layer. This induction of inflammation is implicated in the subsequent breakdown of the endothelial barrier [30].

Astrocytes, integral to various brain regions, also play a crucial role within the neurovascular unit, essential for maintaining BBB integrity. The inflammatory stimulation of cultured astrocytes using IL-1β or TNFα initiated a temporary surge in the production of inflammatory mediators, including IL-6 and TNFα, as well as the anti-inflammatory cytokine IL-13. This response precipitated cell death within a 48 h time period [31].

Inflammatory cytokines can disrupt BBB integrity through multiple mechanisms, including (i) modulating the expression of TJ proteins, (ii) altering the morphological properties of endothelial cells and inducing death, (iii) orchestrating the release of inflammatory cytokines, (iv) initiating the hypoxia-angiogenesis program within endothelial cells, and (v) compromising the supportive functions of astrocytes that are integral to the BBB structure (Figure 4).

In prion diseases, the central nervous system becomes an inflammatory environment due to neuronal and glial cells releasing pro-inflammatory cytokines. This disruption may lead to a feedback loop, exacerbating neuroinflammation by allowing the uncontrolled trafficking of immune cells to the brain. The cytokines cause a breakdown of the BBB’s integrity and function, facilitate unregulated immune cell migration into the brain and create a cycle of increasing inflammation.

## 3. Cytokine Profiles

As research increasingly highlights neuroinflammation’s role in the pathogenesis of TSEs, concerted efforts are underway to elucidate and characterize the cytokine release profile throughout the disease’s progression, potentially serving as an indicator of disease advancement. However, the challenge lies in the heterogeneity of prion strains, which exhibit variations in the incubation periods, clinical manifestations, tissue tropism, and species susceptibility [75,76]. Prion strains and host-specific factors regulate astrocyte response during prion disease showing a region-specific pattern of astrogliosis varying between brain regions [17,77], influencing reactive astrocyte phenotypes in prion diseases [78], and indicating heterogeneity in astrocyte response. At the cellular level, prion strains demonstrate diversity in the regional distribution of prion-induced vacuolar neuropathy and the deposition of PrPSc. Additionally, they show a distinct ‘preference’ for the association of PrPSc with specific brain cell types [79]. The diversity among TSEs adds layers of complexity to the diseases and presents obstacles in establishing a uniform cytokine profile or any other singular marker for tracking disease progression. Nonetheless, understanding these differences is crucial for classifying and studying prion strains.

### Cytokines and Prion Strains

An in vivo study comparing prion strains RML, 22L, and ME7 highlighted strain-specific patterns of PrPSc accumulation in various brain regions and among different cell types, including neurons and astroglia. This differential deposition was observed at early, preclinical stages, ranging from 40 to 100 days post-infection (dpi) [79]. Interestingly, despite their differences, the three strains promoted a similar upregulation of specific inflammatory genes in the thalamus at the mRNA level, including IP-10, Ccl2, Ccl4, Ccl5, Ccl7, and Ccl8. This upregulation corresponded with the initiation of microglia and astrocyte activation. The consistency of this inflammatory gene expression profile across prion strains contrasts with the distinct gene sets triggered by acute viral infections, indicating a potential unique cytokine signature associated with prion diseases [79].

When intracerebrally inoculated into C57BL/6 mice, prion strains ME7 and 139A, as well as the SMB-15 cell line that perpetuates PrPSc replication through cell passage, precipitated a significant increase in the expression of cytokines IP-10, KC, and M-CSF in brains at the terminal stage of the disease [72]. This study supports our hypothesis that various prion strains can invoke a consistent response in cytokine activation, which varies depending on the infection timeline. During the pre-clinical phase, RML, 22L, and ME7 prion strains appear to upregulate a shared suite of inflammatory cytokines—namely IP-10, CCL2, CCL4, CCL5, CCL7, and CCL8—involved with the onset of astroglia activation and chemoattraction of immune cells [38,79,80]. At the terminal stage of the disease, prion strains 139A and ME7 show an upregulation of a distinct cytokine array, predominantly IP-10, KC, and M-CSF [81,82,83] (Figure 5). Therefore, cytokines IP-10, KC, and M-CSF enhance the recruitment of immune cells to the brain which may align with the BBB compromise, further exacerbating neuroinflammation at advanced stages of TSEs (Figure 5).

The correlation between the progression of the disease and the specific cytokines activated was further investigated for the 22L prion strain. Notably, the pro-inflammatory cytokine IL-12p40 was upregulated in brain homogenates from infected mice post-60 days of infection (dpi), indicating a progression in cytokine response despite the onset of significant gliosis already evident at 40 dpi [32]. Together, Carroll et al. 2019 and Tribouillard-Tanvier et al. 2012 suggest that during the pre-clinical phase of the 22L prion strain infection, there is an observable pattern of pro-inflammatory cytokine modulation at the transcript level. Specifically, there is a noted upregulation of mRNA for cytokines such as IP-10, CCL2, CCL4, CCL5, CCl7, and CCL8 [59]. This phase is further characterized by the initial detection of the pro-inflammatory cytokine IL-12p40 at 60 dpi, a subsequent increase in the levels of CCL3, IL-1β, and CXCL1 by 80 dpi, and a rise in CCL2 and CCL5 by 115 days [32]. Notably, there is a thirty-fold increase in the expression of IL-12p40 in the terminal stages when compared to the pre-clinical phase. However, this cytokine’s elevation does not seem to play a pivotal role in the 22L-induced neuroinflammatory and neurodegenerative processes. Evidence from gene knockout studies reveals that the absence of the gene encoding IL-12p40 does not significantly impact the pathological trajectory of the disease [32]. This suggests that the cytokine release cascade, regulated in accordance with the stage of the disease, may be initiated by alternative cytokines or molecules distinct from IL-12p40.

Persistent neuroinflammation is implicated in neuronal death within prion diseases, and the absence of anti-inflammatory mediators could hasten TSE progression. Illustrating this, research using IL-10 knockout mice that were infected with the RML or ME7.1 strains demonstrated that the lack of this anti-inflammatory cytokine accelerated disease progression. Notably, IL-10−/− mice reached the terminal stage of the disease in only about 30% of the time it took for their wild-type counterparts to develop terminal illness [37]. This was due to an early expression of the pro-inflammatory cytokine TNF-α, which appears to sensitize mice to prion-induced pathology [37].

TNF-α was elevated in the cerebrospinal fluid (CSF) of five patients diagnosed with a sporadic or new-variant CJD [84]. The inhibition of the TNF-α signaling pathway led to delayed disease onset following an intraperitoneal injection of the ME7 strain in mice [85]. Indeed, the systemic administration of TNF-α increased cognitive deficits, neuroinflammatory response, and sickness response in ME7-infected animals [86]. The significance of TNF-α varies with the route of infection. TNF-α−/− mice were susceptible to the intracerebral inoculation of the ME7 strain, suggesting TNF-α is not indispensable in establishing the disease [66]. However, the intraperitoneal injection of the ME7 strain in TNF-α−/− mice failed to induce the disease in five out of the eight mice [66]. The observed effects indicate that during infection with prion strains, the pro-inflammatory cytokine TNF-α plays a role in the disease progression, although further investigation is necessary to understand its relevance for disease establishment and progression.

IL-1β also plays a role in prion disease. It was found elevated in the CSF of CJD patients together with TNF-α [84]. PrP recombinant fibrils activate microglia, inducing IL-1β secretion [87]. In mice infected with the 139A strain, the knockout of the IL-1β cytokine receptor (IL1R1−/−) resulted in delayed disease onset and an extension of survival time [88]. Additionally, IL1R1−/− mice showed subdued astrocytic activation at the asymptomatic stage and delayed PrPSc deposition [88]. These findings suggest that an increase in IL-1β release within the brain might precede the accumulation of PrPSc in the 139A strain, indicating a unique aspect of this strain’s pathogenesis. In contrast, mice infected with the ME7 strain showed no detection of IL-1β [89]. Opposing data may differ due to variations in prion strains, techniques used, and neuroinflammation, whether acute or chronic. There is still an opening filed to investigate this and other cytokines as causes or consequences contributing to prion disease.

## 4. Diagnostic and Therapeutic Potential

Despite the complexity of TSEs, emerging research suggests that there is a distinct cytokine profile elicited by prion infection, markedly different from the response seen in viral infections [79]. Various prion strains have been shown to initiate a consistent array of inflammatory cytokines, with the modulation of this response appearing to be tightly coupled with the stage of disease progression [79] (Figure 5). Therefore, a comprehensive understanding of the cytokine release profile could serve as a valuable biomarker for identifying different stages of TSEs. Furthermore, considering the role of neuroinflammation in the pathogenesis of TSEs, therapeutic strategies that target specific cytokines and their signaling pathways may offer new avenues for treatment. An improved grasp of cytokine signaling within the TSEs context could significantly enhance diagnostic and therapeutic approaches.

### Immunomodulation

Immunomodulatory therapies that target cytokine signaling to mitigate the neuroinflammatory response and slow down the progression of prion diseases have been explored and emerged as a promising strategy. An example is the cluster of differentiation 40 ligand (CD40L), a TNF-α homolog that is transiently expressed on activated CD4 T cells and is also present in astrocytes, endothelial cells, and vascular smooth muscle cells [90]. The knockout of CD40L in prion-infected mice led to a rapid disease development, with increased activation of microglia and vacuolation, culminating in animal death 40 days earlier than wild-type infected mice [90]. Therefore, CD40–CD40L interaction appears to be relevant to the pathogenesis of prion diseases, and its stimulation may be neuroprotective. The expression of CD40 was shown to be upregulated in astrocytes exposed to pro-inflammatory cytokines [91], which may represent a cellular strategy to survive in the face of neuroinflammation. Indeed, the modulation of CD40L expression has already been explored in clinical trials of immunotherapy for diseases such as cancer and lupus [92,93], reinforcing its safeness and therapeutic potential for prion diseases.

The type 1 interleukin-1 receptor (IL-1R1) presents a potential target for immunotherapies in prion diseases. Although there are currently no clinical trials targeting its modulation or expression, evidence from IL-1R1 knockout mice showed a significant delay in PrPSc accumulation and disease onset. These mice exhibited an incubation time 16% longer than their wild-type mice counterparts, as well as prolonged survival [88,94]. Therefore, inhibiting IL-1R1 signaling could represent an improvement in patient life. IL-1R1 null mice subjected to brain injury also showed a significant decrease in neuroinflammation, with diminished microglial activation, deficient recruitment of peripheral macrophages, abated astroglia response, and attenuated levels of mRNAs for pro-inflammatory cytokines [95]. In a transgenic mouse model of genetic CJD, antagonizing IL1R1 with anakinra (already used clinically to treat rheumatoid arthritis) normalized hippocampal neurotransmission and reduced seizure susceptibility [96], reinforcing that modulation of IL1R1 signaling or its expression could offer therapeutic benefits to TSEs [97].

Cyclooxygenase-2 (COX-2), predominantly found in inflammatory cells, plays a critical role in the biosynthesis of prostaglandins (PGs). These lipid mediators are essential in regulating inflammatory response. Neuroinflammatory processes involving an increased expression of COX-2 and elevated prostaglandin E2 (PGE2) levels have been associated with several neurodegenerative diseases, such as Parkinson’s disease, Alzheimer’s disease, and amyotrophic lateral sclerosis [98,99]. In prion-infected C57BL/6J mice, COX-2 expression increased with the progression of the disease, being specifically localized to microglia [100]. Furthermore, in a 301V/VM murine scrapie model, PGE2 expression was induced in specific brain regions that exhibited PrPSc accumulation [101]. These studies suggested that COX-2-mediated neuroinflammation is relevant to TSE pathogenesis. For instance, when human neuroblastoma SH-SY5Y cells were exposed to synthetic beta-amyloid peptides, there was an induction of COX-2 expression [99], hinting at the possibility amyloidogenic protein aggregates may trigger COX-2 upregulation, thereby contributing to neurodegeneration. Moreover, since oligodendrocytes with increased COX-2 expression have shown heightened susceptibility to excitotoxic death [102], the upregulation of this enzyme observed during the progression of prion diseases [103] might be linked to neuronal death. Hence, it is plausible to suggest that COX-2 inhibitors could represent promising therapeutical agents for TSEs. However, using existing COX inhibitors in experimental models for neurodegenerative diseases has not yielded encouraging results [104].

The pro-inflammatory cytokine TNF-α is also central in the pathogenesis of prion diseases. Interestingly, SMB-S15 cells, which harbor a persistent prion infection, exhibit heightened susceptibility to TNF-α-induced death compared to their non-infected counterparts, the SMB-PS cells [105]. When exposed to a conditioned medium from LPS-activated microglia, rich in TNF-α, the viability of SMB-S15 cells was significantly reduced, whereas the SMB-PS cell remained unaffected [105]. This indicates that TNF-α exerts a modulatory effect in prion-infected cells distinctly from healthy ones under similar conditions. The increased susceptibility of infected cells could be due to the accumulation of misfolded and possibly dysfunctional prion proteins, since the cellular prion PrPC was shown to be protective against TNF-α-mediated inflammation [106,107]. The selective cytotoxicity of TNF-α on prion-infected cells presents a potential therapeutical perspective for TSEs. Indeed, TNF-α antagonists have been used clinically for over two decades across various inflammatory diseases and have become the cornerstone of immunotherapy for such conditions [108]. Recently, inhibiting TNF-α signaling has been explored in a phase I clinical trial with stage IV lung adenocarcinoma patients, showing promising results [109]. Certolizumab, an anti-TNF-α monoclonal antibody, led to the diminishment of peripheral cytokines associated with the paracrine inflammatory loop induced by chemotherapy [109].

As a part of the neuroinflammation cascade, reactive astrocytes release ROS, which may lead to oxidative stress and neuronal damage [110,111,112]. The activation of the Nrf2/ARE pathway has been shown to reduce neuroinflammation by upregulating the expression of antioxidant enzymes and decreasing oxidative stress [91]. Nrf2 activity declines with age, making the brain more susceptible to oxidative injury, abnormal protein aggregation, and neurodegeneration [113]. The activation of Nrf2 was shown to be neuroprotective in both in vitro and in vivo models for neurodegenerative diseases, which suggests a particular relevance for this pathway in these disorders [114,115]. The antioxidant properties of carnosic acid and carnosol, two compounds isolated from rosemary leaves, involve the upregulation of genes related to cellular redox response, including the gene encoding for Nrf2 and NFE2L2 [116]. Both compounds were found to inhibit the formation of PrPSc in N2a22L cells, which harbor a persistent prion infection, and in a cell-free system [116]. The observed upregulation of Nrf2 by these compounds indicates a multifaceted mechanism of action that includes anti-prion and disaggregase activities. These properties present promising therapeutic potential for the treatment of TSEs.

Pro-inflammatory cytokine IL-1 has been shown to promote multiple deleterious effects within the brain and to stimulate neuroinflammation [22,31]. In a coculture system, microglial IL-1 α and IL-1β, along with TNF-α, induced the phosphorylation of tau protein, which makes it more prone to aggregate. This was accompanied by a decrease in synaptophysin expression in primary neurons, leading to a reduction in the number of synapses [117]. IL-1 receptor antagonist (IL-1Ra) and anti-IL-1β antibody, but not anti-TNF-α antibody, attenuated the effect of activated microglia on neuronal tau and synaptophysin [117]. IL-1Ra is produced endogenously by hepatic cells and can cross the BBB [118]. Therefore, stimulating the synthesis of endogenous IL-1Ra could be a promising strategy to reduce neuroinflammation. This is because IL-1Ra can reach the brain, which could be therapeutically valuable for TSEs. Indeed, the administration of IL-1Ra has been proposed as a therapy and explored clinically in sepsis and rheumatoid arthritis (RA), promoting positive clinical effects in RA [119]. Targeting IL-1Ra is an interesting approach for immunotherapy development since inhibiting a single mediator of the immune response may have a clinical impact.

## 5. Concluding Remarks

The intricate web of interactions between prions, the immune system, and neuroinflammatory processes highlights the complex nature of TSE pathogenesis. The discovery of distinct cytokine profiles associated with different prion strains has opened new avenues for understanding disease progression. It is evident that both temporal and spatial patterns of cytokine release are intricately linked with disease phases. Notably, certain pro-inflammatory cytokines, such as TNF-α, and their receptors, like IL-1R1, have been implicated in accelerating TSE progression, indicating potential targets for therapeutic intervention.

Furthermore, the nuances of glial and neuronal responses in the presence of prion infection point towards the possibility of exploiting these cellular mechanisms for the development of diagnostics and treatments. For instance, the application of immunomodulatory strategies utilizing COX-2 inhibitors or TNF-α antagonists, although yet to yield conclusive benefits, remains a promising field of exploration.

The studies reviewed here suggest that the role of neuroinflammation in TSEs is not merely a bystander effect but a critical element that could be harnessed for therapeutic gain. Future research should continue to dissect the contributions of individual cytokines and the molecular pathways they engage in. This effort will be crucial in crafting interventions that can delicately balance the immune system’s response—mitigating detrimental inflammation without compromising innate protective mechanisms.

As we move forward, the translational potential of these findings beckons a cautious yet optimistic approach. Given the complexity of cytokine interactions and the brain’s immune environment, a multifaceted strategy encompassing immunomodulation, the regulation of the BBB, and targeted delivery systems could eventually culminate in effective therapies for TSEs. Continued meticulous research is required to unravel the precise roles of these elements. Yet, the advances in our understanding offer a beacon of hope. Through continued research, we edge closer to unveiling novel therapeutics that may one day transform the prognosis of TSEs and similar neurodegenerative diseases.

## Figures and Tables

**Figure 1 brainsci-14-00413-f001:**
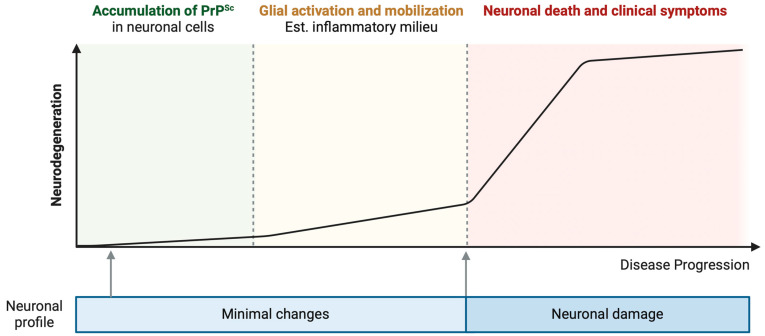
Progression of neurodegenerative disease correlated with PrPSc accumulation, glial activation, and neuronal loss. The hypothesis illustrated in this figure posits a sequential progression in prion disease initiating with a subclinical phase, during which PrPSc accumulates within neuronal cells, eliciting minimal alterations in their morphology and function [13,15]. This period is succeeded by a phase distinguished by the activation of glial cells and the development of an inflammatory environment, preceding pronounced neurodegeneration [13]. Notably, this phase may progress without overt clinical manifestations [13]. The subsequent stage is marked by accelerated neurodegeneration, culminating in significant neuronal death and the emergence of clinical symptoms [13]. Created with BioRender.com (accessed on 16 April 2024).

**Figure 2 brainsci-14-00413-f002:**
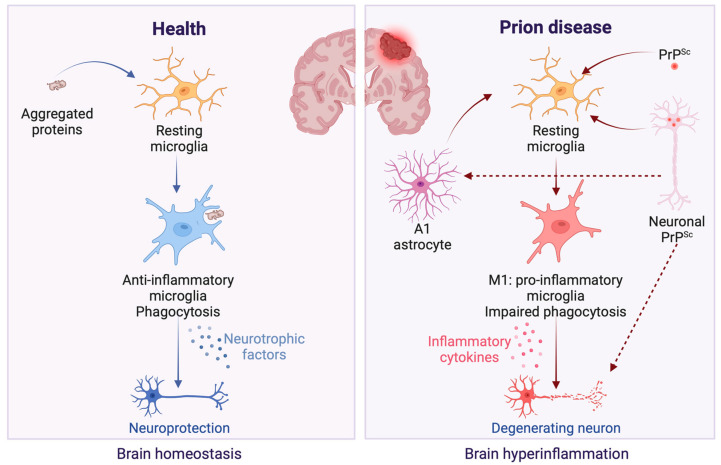
Microglial response in healthy versus prion disease conditions in the brain. In the healthy brain, microglial cells exert neuroprotective effects, as depicted on the figure’s left panel [43]. These effects are mediated through the efficient clearance of protein aggregates via phagocytosis and the secretion of neurotrophic factors, which collectively contribute to preserving neuronal structure and functionality [43]. In contrast, the presence of PrPSc in prion diseases initiates a detrimental inflammatory response, illustrated in the figure’s right panel. Neuronal PrPSc aggregates prompt microglia to adopt an M1 phenotype [11,25], fostering an inflammatory milieu detrimental to neuronal health. This state is characterized by diminished phagocytic activity and an increased release of pro-inflammatory cytokines [39], which, in turn, facilitate the conversion of astrocytes to A1 phenotype that supports further M1 microglial activation [22,24,25], exacerbating the cycle of neuroinflammation and leading to progressive neuronal damage [39]. Created with BioRender.com (accessed on 16 April 2024).

**Figure 4 brainsci-14-00413-f004:**
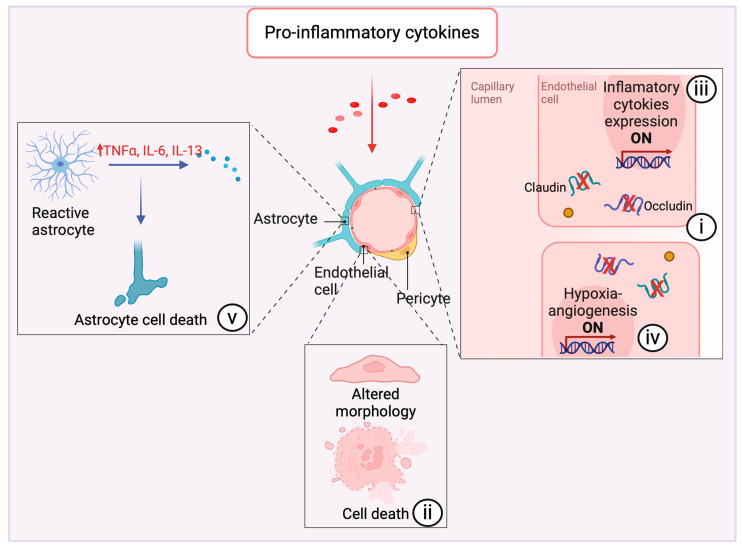
Impact of proinflammatory cytokines on neurovascular integrity and astrocyte function. Pro-inflammatory cytokines exert multifaceted effects on the cerebrovascular system, precipitating a cascade of alterations detrimental to the integrity of the neurovascular unit. Notably, these cytokines can induce (i) downregulation of integral tight junction proteins such as claudin and occludin [29,58,64,65,70]. This regulatory shift compromises endothelial barrier function, manifesting as enhanced vascular permeability. (ii) Morphological changes in endothelial cells are observed under inflammatory stress, and they can undergo necroptosis as a direct consequence of prolonged exposure to pro-inflammatory cytokines [72,73]. (iii) These cytokines activate gene expression pathways within endothelial cells that encode for additional pro-inflammatory mediators [30], thus establishing a self-propagating loop of inflammation, and (iv) augments the hypoxia-angiogenesis response [74]. (v) Astrocytes assume a reactive state in response to cytokine exposure and begin secreting pro-inflammatory factors [31]. This reactive astrocytosis leads to astrocyte cell death [31], further intensifying the inflammatory milieu. Created with BioRender.com (accessed on 16 April 2024).

**Figure 5 brainsci-14-00413-f005:**
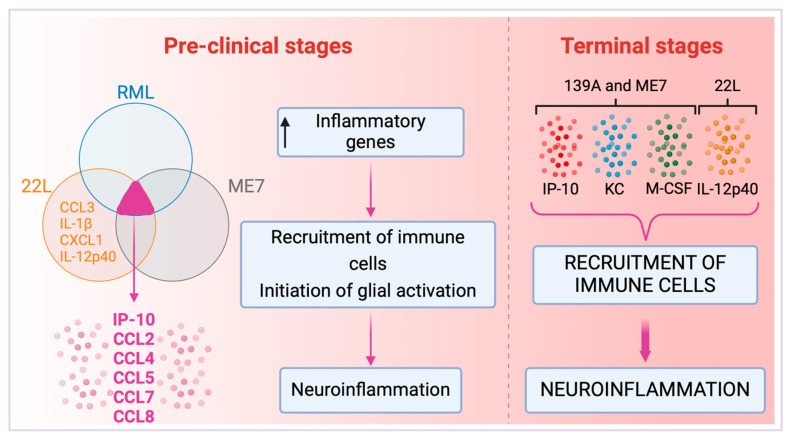
Cytokine profiles in pre-clinical and terminal stages induced by prion strains. Comparative overview of cytokine profiles at different stages of prion diseases induced by strains RML, 22L, ME7, and 139A. The Venn diagram illustrates the shared inflammatory cytokines upregulated by the RML, 22L, and ME7 strains, indicating a typical early-stage inflammatory response in the pre-clinical phase of the disease [79]. Key cytokines are highlighted within the overlapping section, which signifies the unified response across the three prion strains [79]. The 22L strain, upon further investigation, revealed an expanded spectrum of cytokines (CCL3, IL-1β, CXCL1, and IL-12p40) present in the pre-clinical stage [32,59]. This collective upregulation of inflammatory genes results in the recruitment of immune cells and initiating glial activation, setting the stage for progressive neuroinflammation. At the terminal phase of the disease, the 139A strain, evaluated solely at this stage, alongside 22L and ME7 strains, demonstrates a novel repertoire of cytokines, including IP-10, KC, M-CSF, and IL-12p40 [32], differing from those in the pre-clinical stage, and are indicative of a more advanced inflammatory state. The concentration of the cytokine 12p40 was markedly increased, exhibiting a thirty-fold elevation in the terminal stage relative to the pre-clinical stage for the 22L strain [32]. The progression to the terminal stage is characterized by increased recruitment of immune cells and exacerbated neuroinflammation. Notably, the RML strain was not evaluated at the terminal stage. Created with BioRender.com (accessed on 16 April 2024).

**Table 1 brainsci-14-00413-t001:** Cytokines in neuroinflammation: sources and functions.

Cytokine	Source	Pro-Inflammatory or Anti-Inflammatory?	Reference
IL-10	Activated astrocytes and microglia; neurons	Anti-inflammatory	[21]
IL-1α	Activated microglia	Pro-inflammatory	[22]
TNF-α	Activated microglia	Pro-inflammatory	[22]
	Neurons	Anti-inflammatory	[23]
C1q	Activated microglia	Pro-inflammatory	[22]
CCL2	Reactive A1 astrocytes	Pro-inflammatory	[24,25]
IL-3	Neurons	Anti-inflammatory	[23]
IL-4	Neurons	Anti-inflammatory	[23]
IL-6	Neurons; astrocytes	Pro-inflammatory	[26,27]
MCP-1	Neurons	Pro-inflammatory	[28]
IL-17A	Th17 lymphocytes	Pro-inflammatory	[29]
IL-22	Th17 lymphocytes	Pro-inflammatory	[29]
IL-8	endothelial cell layer	Pro-inflammatory	[30]
IL-1	Activated astrocytes	Pro-inflammatory	[30]
IL-13	Activated astrocytes	Anti-inflammatory	[31]
IL-12p40	Brain homogenates (22L-strain-infected mice)	Pro-inflammatory	[32] *

* This study quantified cytokine release in whole-brain homogenates without identifying the specific cell types responsible for production.

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
