# Peer review of "Interactions between Cytokines and the Pathogenesis of Prion Diseases: Insights and Implications"

_brainsci, 2024, doi:10.3390/brainsci14050413_

Round 1
Reviewer 1 Report
Comments and Suggestions for Authors
This manuscript presents a most intriguing and impressive review examining the role of cytokines in elucidating the pathogenesis of prion diseases. I am most grateful for the opportunity to review it further.
However, there are some rather serious concerns with the manuscript that I would urge be addressed:
1. The pathological model of prion disease depicted in Figure 1 appears to be overly inferential, lacking sufficient evidential basis. Firstly, does the pathogenesis truly commence with the aggregation of abnormal prion proteins in neurons? This remains unclear. One would expect abnormal prion protein deposition to occur not solely in neurons, but also in microglia. Does this happen concomitantly? Do neurons or microglia exhibit changes first? This critical point requires clarification using animal models before we can properly proceed with the manuscript.
2. Moreover, while neuroinflammation is principally focused on microglial activity within the glial system, the relationship to astrocytes is inadequately characterised. Furthermore, astrocytes and pericytes collectively ensheath blood vessels forming the blood-brain barrier. The roles of these two glial cell types are not delineated in this review. I would strongly recommend explicitly detailing the nature of this relationship within the manuscript.
Comments on the Quality of English LanguageNo issue with the quality of English per se, but rather the manner in which the content of the manuscript is presented leaves much to be desired.
Author Response
Thank you for providing valuable feedback on the manuscript “Interactions Between Cytokines and the Pathogenesis of Prion Diseases: Insights and Implications”. We appreciate your time and effort dedicated to evaluating our work. Your comments and suggestions have been instrumental in enhancing the quality of the manuscript.
Please find below a detailed point-by-point response. To facilitate the revision process, all changes made in the new version of the manuscript are highlighted in yellow.
Comments 1: The pathological model of prion disease depicted in Figure 1 appears to be overly inferential, lacking sufficient evidential basis. Firstly, does the pathogenesis truly commence with the aggregation of abnormal prion proteins in neurons? This remains unclear. One would expect abnormal prion protein deposition to occur not solely in neurons, but also in microglia. Does this happen concomitantly? Do neurons or microglia exhibit changes first? This critical point requires clarification using animal models before we can properly proceed with the manuscript.
Response 1: Thank you for pointing these questions out. In recognition of the current literature, we acknowledge that the data available are still limited in describing the full breadth of events depicted in Figure 1. However, these findings are instrumental in formulating a hypothesis. To this end, we have emphasized in the legend of Figure 1 that the illustration represents a hypothesized model (lines 117-118). Each finding described in the legend is now accompanied by reference numbers corresponding to the literature that showed these specific observations.
We concur that the data to definitively establish the initiation of pathogenesis in prion diseases is still scant. However, pertinent to your query, Lakkaraju's 2022 study (ref 11) provided insights suggesting that PrPSc accumulation in astrocytes does not induce microgliosis or astrogliosis, nor does it precipitate the clinical onset of disease in animal models. Contrastingly, PrPSc propagation within neurons was correlated with disease development (lines 91-95). We altered the text to make this information clear (Lines 99-100). Whether PrPSc accumulation occurs simultaneously in neurons and glial cells during disease progression remains to be elucidated.
Regarding microglia, the presence or absence of the cellular prion protein (PrPC) — which would serve as a substrate for PrPSc propagation — is a matter of ongoing debate. For instance, Pinheiro's 2015 study showed that PrPC is not essential for LPS-induced microglia activation, whereas Peggion's 2020 study presented opposing evidence. Peggion raises the hypothesis that differences in the genetic background of the knockout animals may explain this discrepancy. We have included this nuanced information in our revised text to reflect these diverse findings and hypotheses (lines 100-104).
Comments 2: Moreover, while neuroinflammation is principally focused on microglial activity within the glial system, the relationship to astrocytes is inadequately characterised. Furthermore, astrocytes and pericytes collectively ensheath blood vessels forming the blood-brain barrier. The roles of these two glial cell types are not delineated in this review. I would strongly recommend explicitly detailing the nature of this relationship within the manuscript.
Response 2: We have revised the text to clarify the importance of PrPSc propagation in astrocytes (lines 99-103) and aim to enhance understanding of this commentary. Additionally, the role of astrocytes, particularly in the context of the blood-brain barrier, is discussed in Section 2.3, highlighting their contribution to maintaining its integrity. The release of cytokines upon glial activation, involving both microglia and astrocytes, is addressed throughout Section 3, indicating their synergistic roles in prion neuroinflammation.
Comments on the Quality of English Language
Point: No issue with the quality of English per se, but rather the manner in which the content of the manuscript is presented leaves much to be desired.
Response: Thank you for your feedback regarding the content presentation within our manuscript. We acknowledge that while the language quality is satisfactory, the organization and clarity of the information conveyed may require further refinement for better comprehension and impact. To address this, we have made specific adjustments to ensure that the sequence of arguments is logical, and the overall narrative is engaging and informative.
Reviewer 2 Report
Comments and Suggestions for Authors
The manuscript for review by Assis-de-Lemos et al. investigates the “Interactions Between Cytokines and the Pathogenesis of Prion Diseases”. The paper summarises well the current knowledge in the field, however I do have several minor comments below which should be addressed.
Section 2. Immune response activation in prion infection. This entire section/statement should be qualified studies suggesting microglia phagocytose and degrade prions observed an enhanced accumulation of prions under microglia deletion conditions 1,2. Studies using microglia deficient mice observed no difference in prion accumulation3. Furthermore these studies also revealed a shortening of disease survival time in the absence of microglial-derived inflammatory responses and therefore conclude their overall influence is neuroprotective. These findings are supported by other studies that confirm neuronal PrPC and its conversion into prions is required for neurotoxicity.
1. References throughout the manuscript to TNFα and Il-6 should be qualified in light of the fact their absence had no impact on CNS infection with ME7 scrapie
2. in section 2.1 The reviewer would argue that astrocyte activation is one of the most obvious and common features of prion diseases throughout human and animal host species and prion strain variations. Of relevance to later comments and figure 1 in many experimental prion strains the distribution of prion accumulation is predominantly associated with astrocytes not neurons – see figure 10.
3. In line 123 the relative incubation period in prion disease is not an indication of relative susceptibility, these are independent variables. IL-10 deficiency accelerate prion disease incubation period however limiting dilution data is not presented in the corresponding reference that allows the authors to confirm any statement regards relative susceptibility of these mice to prion disease. This statement should be corrected accordingly.
4. In line 127 data from several studies shows that A1 astrocytes are not found in prion disease – prion activated astrocytes display a mixture of A1 and A2 genes described by Liddelow et al., this statement should also be qualified in light of the observation that astrocyte activation precedes in the complete absence of microglia3
5. Line 201. Further review and evidence regarding the impact of peripheral infection on glial responses are summarised in
6. Lines 312-320 strain and host specific factors also regulate astrocyte response during prion disease with a greater degree of specificity than is observed in vacuolation or prion distribution within the CNS perhaps of greater relevance to this current review.
7. In line 322 in addition to previous work which has already characterised prion deposition over a greater number of strains
1 Zhu, C. et al. A neuroprotective role for microglia during prion diseaes. Journal of Experimental Medicine 213, 1047-1059 (2016).
2 Carroll, J. A., Race, B., Williams, K., Striebel, J. & Chesebro, B. Microglia Are Critical in Host Defense against Prion Disease. J Virol 92, e00549-00518, doi:10.1128/JVI.00549-18 (2018).
3 Bradford, B. M., McGuire, L. I., Hume, D. A., Pridans, C. & Mabbott, N. A. Microglia deficiency accelerates prion disease but does not enhance prion accumulation in the brain. Glia 70, 2169-2187, doi:10.1002/glia.24244 (2022).
Author Response
Thank you for providing valuable feedback on the manuscript “Interactions Between Cytokines and the Pathogenesis of Prion Diseases: Insights and Implications”. We appreciate your time and effort dedicated to evaluating our work. Your comments and suggestions have been instrumental in enhancing the quality of the manuscript.
Please find below a detailed point-by-point response. To facilitate the revision process, all changes made in the new version of the manuscript are highlighted in yellow.
Comments 1: Section 2. Immune response activation in prion infection. This entire section/statement should be qualified studies suggesting microglia phagocytose and degrade prions observed an enhanced accumulation of prions under microglia deletion conditions 1,2. Studies using microglia deficient mice observed no difference in prion accumulation3. Furthermore these studies also revealed a shortening of disease survival time in the absence of microglial-derived inflammatory responses and therefore conclude their overall influence is neuroprotective. These findings are supported by other studies that confirm neuronal PrPC and its conversion into prions is required for neurotoxicity 4,5.
Response 1: We greatly appreciate your insightful comments and the references you have cited in your review. We have noticed that the correlation between the reference numbers and their respective articles was omitted. We have reviewed our manuscript and bibliography to ensure the suggested references are included and accurately represented. We hope that our revised manuscript now fully reflects and acknowledges the contributions of these references to our work.
Comments 2: References throughout the manuscript to TNFα and Il-6 should be qualified in light of the fact their absence had no impact on CNS infection with ME7 scrapie 6.
Response 2: Agree. We have included and discussed the data of other references about TNFα (lines 425-436) and IL-6 (line 277).
Comments 3: in section 2.1 The reviewer would argue that astrocyte activation is one of the most obvious and common features of prion diseases throughout human and animal host species and prion strain variations. Of relevance to later comments and figure 1 in many experimental prion strains the distribution of prion accumulation is predominantly associated with astrocytes not neurons – see figure 10 7.
Response 3: Pertinent to your query, Lakkaraju's 2022 study (ref 11) provided insights suggesting that PrPSc accumulation in astrocytes does not induce microgliosis or astrogliosis, nor does it precipitate the clinical onset of disease in animal models. Contrastingly, PrPSc propagation within neurons was correlated with disease development (lines 91-95). We altered the text to make this information clear (Lines 99-100). Whether PrPSc accumulation occurs simultaneously in neurons and glial cells during disease progression remains to be elucidated.
Comments 4: In line 123 the relative incubation period in prion disease is not an indication of relative susceptibility, these are independent variables. IL-10 deficiency accelerate prion disease incubation period however limiting dilution data is not presented in the corresponding reference that allows the authors to confirm any statement regards relative susceptibility of these mice to prion disease. This statement should be corrected accordingly.
Response 4: Thank you for bringing that to our attention. We corrected (line 142).
Comments 5: In line 127 data from several studies shows that A1 astrocytes are not found in prion disease – prion activated astrocytes display a mixture of A1 and A2 genes described by Liddelow et al., this statement should also be qualified in light of the observation that astrocyte activation precedes in the complete absence of microglia3
Response 5: Thank you for your insightful comments regarding the role of A1 astrocytes in prion diseases and the associated mechanisms of astrocyte activation. We have carefully considered the studies you referenced and have revised our text to better reflect the complexity and nuances of astrocyte phenotypes in the context of prion pathologies (lines 156-166).
Our revised manuscript now includes a discussion of Hartmann et al.'s findings, which demonstrate the prevalence of A1 astrocytes (C3+-PrPSc-reactive-astrocytes, a subtype of A1-astrocytes) in prion diseases and the upregulation of A1 astrocytes observed by Ugalde et al. in brains affected by sporadic Creutzfeldt-Jakob disease. Moreover, we have incorporated the perspective offered by Liddelow et al., which postulates the presence of A1 astrocytes in major neurodegenerative diseases.
We also acknowledge that Hartmann et al. found that the removal of microglia-derived factors known to induce the A1 state—TNF-α, IL-1α, and C1qa—had minimal impact on suppressing A1-specific markers in prion-infected animals, which, unexpectedly, hastened the progression of prion diseases. Our revised text addresses this finding, highlighting the possibility that astrocyte activation in prion diseases may occur independently of microglia.
Furthermore, we've added information from Carrol et al. and Makarava et al. on the increase of A2 astrocytes in prion-infected mice, suggesting a simultaneous neurotoxic and neuroprotective response during infection and the subsequent dysregulation of these responses concerning neurodegeneration.
Our revisions aim to clarify these points and present a more nuanced view of the interplay between astrocytic activation and prion diseases.
Comments 6: Line 201. Further review and evidence regarding the impact of peripheral infection on glial responses are summarised in 8.
Response 6: Thank you for your suggestion to include reference 8. Unfortunately, the full citation details for this reference were not provided, and we were unable to incorporate it into our text.
Comments 7: Lines 312-320 strain and host specific factors also regulate astrocyte response during prion disease with a greater degree of specificity than is observed in vacuolation or prion distribution within the CNS 7 perhaps of greater relevance to this current review.
Response 7: Thank you for your suggestion to include reference 7. Unfortunately, the full citation details for this reference were not provided. We incorporated your suggestion, including other references (lines 346-349).
Comments 8: In line 322 in addition to previous work which has already characterised prion deposition over a greater number of strains 9.
Response 8: Thank you for your suggestion to include reference 9. Unfortunately, the full citation details for this reference were not provided, and we were unable to incorporate it into our text.
Reviewer 3 Report
Comments and Suggestions for Authors
My suggestions:
1. I would add tables in the manuscript, which may show, how different immune molecules are activated in case of prion accumulation and how what pathways they could induce.
2. Are there any drug trials based on immune reactions, which were used in prion diseases? Authors may mention briefly the immune-based drugs, which were used in other diseases, such as Alzheimer's or Parkinson's disease.
3. Some figures may be added in bigger resolutions, such as bigger font size since they are hard to read.
4. Is there any difference in immune activation patterns in the case of familial-sporadic and iatrogenic forms of prion diseases?
5. Is there any difference in immune activation patterns in the case of GSS, FFI, CJD, and other forms of prion diseases?
Author Response
Thank you for providing valuable feedback on the manuscript “Interactions Between Cytokines and the Pathogenesis of Prion Diseases: Insights and Implications”. We appreciate your time and effort dedicated to evaluating our work. Your comments and suggestions have been instrumental in enhancing the quality of the manuscript.
Please find below a detailed point-by-point response. To facilitate the revision process, all changes made in the new version of the manuscript are highlighted in yellow.
Comments 1: I would add tables in the manuscript, which may show, how different immune molecules are activated in case of prion accumulation and how what pathways they could induce.
Response 1: We appreciate the reviewer's suggestion and have incorporated a table that outlines the cytokines discussed in this study, including their sources and activities. We did not limit this table to prion diseases alone, as the literature identifying the specific cells activated and secreting these cytokines in these conditions is relatively sparse.
Comments 2: Are there any drug trials based on immune reactions, which were used in prion diseases? Authors may mention briefly the immune-based drugs, which were used in other diseases, such as Alzheimer's or Parkinson's disease.
Response 2: Various immunomodulatory approaches are being explored as potential treatments for prion diseases. Various forms of immunization, including passive and active, have been considered. A humanized anti-PrPC monoclonal antibody, ICSM18, is currently undergoing human trials. Strategies targeting the innate immune system's interaction with prions are also under investigation. Some studies have focused on cytokine modulation to alter the immune response in prion diseases, the focus of this Review, but in this case, there are/were no trials.
Similar immune-based therapies are being explored for Alzheimer's and Parkinson's diseases. For Alzheimer’s Disease, passive and active immunization against amyloid-beta is being tested, with some emerging data from human trials suggesting the potential for reducing amyloid-related pathology. For Parkinson’s Disease, immunotherapies targeting alpha-synuclein aggregates are under investigation, although these are at an earlier stage compared to those for Alzheimer's disease.
There was a phase I-II parallel design, randomized, double-blind clinical trial using ingested hrIFN-a to prevent deterioration of cognitive functioning in patients with dementia of Alzheimer's type (AD), but it shows an unknown status at ClinicalTrials.gov.
Comments 3: Some figures may be added in bigger resolutions, such as bigger font size since they are hard to read.
Response 3: We increased all font sizes, and the figure’s original files are high-resolution.
Comments 4: Is there any difference in immune activation patterns in the case of familial-sporadic and iatrogenic forms of prion diseases?
Response 4: The immune activation patterns differ between familial, sporadic, and iatrogenic forms of prion diseases. In sporadic Creutzfeldt-Jakob disease (CJD), the extent of microglial activation is influenced by the biochemical type of PrPSc. However, brain microglia activation in fatal familial insomnia and G114V genetic CJD is minimal, suggesting distinct immune response mechanisms in familial and sporadic forms compared to iatrogenic cases. We have made changes to lines 104-109 to emphasize this distinction.
Comments 5: Is there any difference in immune activation patterns in the case of GSS, FFI, CJD, and other forms of prion diseases?
Response 5: Included as stated above.
Round 2
Reviewer 3 Report
Comments and Suggestions for Authors
The authors fulfilled my suggestions.